# Integrated Metabolomics and Metagenomics Unveiled Biomarkers of Antioxidant Potential in Fermented Brewer’s Grains

**DOI:** 10.3390/antiox13070872

**Published:** 2024-07-20

**Authors:** Hammad Qamar, Yuanfei Li, Rong He, Muhammad Waqas, Min Song, Dun Deng, Yiyan Cui, Pan Yang, Zhichang Liu, Bilal Qammar, Muhammad Asnan, Xiangxue Xie, Miao Yu, Xianyong Ma

**Affiliations:** 1Institute of Animal Science, Guangdong Academy of Agricultural Sciences, State Key Laboratory of Livestock and Poultry Breeding, Key Laboratory of Animal Nutrition and Feed Science in South China, Ministry of Agriculture and Rural Affairs, Guangdong Provincial Key Laboratory of Animal Breeding and Nutrition, Guangzhou 510640, China; drhammadqamar@gmail.com (H.Q.); herong4747@163.com (R.H.); songmin@gdaas.cn (M.S.); dengdun@gdaas.cn (D.D.); cuiyiyan@gdaas.cn (Y.C.); yangpan@gdaas.cn (P.Y.); liuzhichang@gdaas.cn (Z.L.); yumiao@gdaas.cn (M.Y.); 2Institute of Biological Technology, Jiangxi Provincial Key Laboratory of Poultry Genetic Improvement, Nanchang Normal University, Nanchang 330032, China; li-yuan-fei@outlook.com; 3Faculty of Veterinary and Animal Sciences, University of Poonch Rawalakot, Rawalakot 12350, Pakistan; muhammadwaqas@upr.edu.pk; 4Shalamar Hospital, Lahore 54000, Pakistan; bilal.qamar.5680@gmail.com; 5Institute of Animal and Dairy Sciences, University of Agriculture, Faisalabad 38000, Pakistan; asnannaeem72@gmail.com; 6Guangdong VTR Bio-Tech Co., Ltd., Zhuhai 519060, China; 7Maoming Branch, Guangdong Laboratory for Lingnan Modern Agricultural, Maoming 525000, China

**Keywords:** antioxidants, fermentation, *Lactobacillus*, enzymes, metagenomics, metabolomics, isoflavonoids, flavonoids, phenylpropanoids, brewer’s spent grain

## Abstract

About one-third of the global food supply is wasted. Brewers’ spent grain (BSG), being produced in enormous amounts by the brewery industry, possesses an eminence nutritional profile, yet its recycling is often neglected for multiple reasons. We employed integrated metagenomics and metabolomics techniques to assess the effects of enzyme treatments and *Lactobacillus* fermentation on the antioxidant capacity of BSG. The biotreated BSG revealed improved antioxidant capability, as evidenced by significantly increased (*p* < 0.05) radical scavenging activity and flavonoid and polyphenol content. Untargeted metabolomics revealed that *Lactobacillus* fermentation led to the prominent synthesis (*p* < 0.05) of 15 novel antioxidant peptides, as well as significantly higher (*p* < 0.05) enrichment of isoflavonoid and phenylpropanoid biosynthesis pathways. The correlation analysis demonstrated that *Lactiplantibacillus plantarum* exhibited strong correlation (*p* < 0.05) with aucubin and carbohydrate-active enzymes, namely, glycoside hydrolases 25, glycosyl transferases 5, and carbohydrate esterases 9. The fermented BSG has potential applications in the food industry as a culture medium, a functional food component for human consumption, and a bioactive feed ingredient for animals.

## 1. Introduction

Food processing results in almost one-third of the global food wastage, thereby creating enormous environmental, economic, and climatic challenges [1]. As the world population grows, food production is expected to escalate, leading to even greater amounts of waste annually. However, the nutritional content of these food processing wastes may be reused for other applications, providing economical and sustainable solutions to the worldwide food waste challenges [2].

Brewers’ spent grain (BSG) accounts for 85% of the total waste produced by the brewing industry [3]. Other waste ingredients include discarded hops and yeast [4]. Globally, about 39 million tons of BSG are generated each year [5]. Because of the challenges involved in preserving these wastes owing to their high moisture percentage and complex composition, excess BSG is likely to be disposed of, leading to environmental pollution, or used as cattle feed, contributing to methane gas production [6]. BSG possesses a high moisture percentage of approximately 70% and is rich in nutrients, making it an ideal environment for microbes to thrive [7]. This contamination is largely caused by filamentous fungus and begins shortly after BSG production. The environmental issues that arise in this situation are substantial due to the huge amount of bacteria growing and multiplying in the environment. Furthermore, each ton of BSG discarded into the environment accounts for greenhouse gas emissions equivalent to 513 kg of CO_2_ [8,9]. Hence, it is essential to explore alternate, cost-effective, and environmentally friendly uses for BSG.

Producing valuable bio-products, including amino acids, antibiotics, vitamins, enzymes, and bulk chemicals, using microorganisms is a prominent focus in the biotech industry [10]. Utilizing biomass from agro-industrial waste for biotech applications is both beneficial and sustainable, as it avoids competing with global food sources [11].

Barley malt grain husks, pericarp, and seed coat layers make up the components of BSG [12,13]. These materials are rich in cellulose, non-cellulosic polysaccharides, and lignin [12]. Arabinoxylans (AXs), the fundamental component of hemicellulose, are present in BSG up to 40% on a dry matter basis and are composed of either a xylose or an arabinose residue as the main skeleton structure [13]. BSG also contains proteins, essential amino acids, fiber, and phenolic compounds [4,7]. Lipids, fatty acids, and polyphenols are also present in BSG [14,15], with triglyceride being the most common lipid and fatty acids like linoleic, palmitic, oleic, α-linoleic, and stearic acids also being identified [14]. Extracting useful components from bioenergy and sustainable growth BSG poses a significant challenge due to the fact that proteins and nutrients are tightly bound to cellulose and hemicellulose [13]. To overcome this obstacle, various methods, such as physical, biological, and chemical pre-treatments, or a combination of these treatments, are necessary [16]. Among these options, biological pre-treatment emerges as a more eco-friendly solution since it does not involve the use of chemicals or solvents, and it also eliminates the risk of producing toxic by-products [17].

The pre-treatment of the enzymes prevents the production of toxins in the final product and results in a product that is better for the environment when used in the treatment of BSG. The commercially available enzymes have been utilized to extract sugars and nutrients from BSG [4,18]. The BSG Axs are composed of notable amounts of water-extractable portion and release xylooligosaccharides upon enzymatic treatment [19]. These nutrients released through enzymatic hydrolysis can then be used for microbial fermentation. At the same time, microbes have the ability to create both primary and secondary compounds, which can then enhance the value of the original BSG even more [20]. This fermentation process has been shown to enhance the crude and soluble protein contents of BSG [21], and also to produce lactic acid [22]. Lactic acid is produced by *Lactobacillus* during carbohydrate fermentation, and this microbe is widely used as a probiotic due to its antioxidant properties [23,24].

The importance of *Lactobacillus* fermentation in boosting the antioxidant potential of various agro-industry by-products is increasingly acknowledged. The fermentation carried out by *Lactobacillus* spp. Results in the liberation and stimulation of antioxidant substances, such as phenolic acids and flavonoids, from plant-derived materials [25]. *Lactobacillus* fermentation boosts antioxidant capacity via enzymatic hydrolysis, generating bioactive chemicals, and stimulating antioxidant enzymes. This process enhances the antioxidants’ bioavailability and decreases substances that hinder the effectiveness of antioxidants, thereby strengthening the total antioxidant capacity of the fermented substance [26].

The overall quality of the fermented food is mostly determined by the microbial composition and their byproducts [27]. At the moment, advanced sequencing analysis, “omics” technologies, and integration of metagenomic and metabolomic techniques are proven to be effective tools for performing deep analyses of bacterial communities and assessing the metabolites of fermented material [28]. Metabolomics allows for the examination of numerous individual compounds at once, rather than in groups. Employing metabolomics to study BSG fermentation proves to be an effective method for evaluating the distinct compounds. These methods appear to be successful in understanding the impact of the entire microbial community and microbial secondary metabolites on fermented food’s nutrient composition and functional roles. They can provide deep insights into the microbial communities’ composition, metabolites’ composition, and their biological roles [29].

Not much attention has been given to decreasing the anti-nutritional elements and enhancing the nutritional benefits of brewer’s spent grain using a combination of enzymatic breakdown and microbial fermentation. This research aims to uncover the metabolic alterations and processes involved in the fermentation of BSG by microbes. This research hypothesizes that breaking down fibers with enzymes can help boost lactobacilli fermentation and enhance the antioxidant and nutritional value of BSG.

## 2. Materials and Methods

### 2.1. Bacterial Microbes and Enzymes

The two commercially available probiotics *Lactobacillus plantarum* and *Lactobacillus reuteri* were procured from the Shandong Hezhong Kangyuan Biotechnology (Shandong, China) and Xian Xihai Biotechnology (Xian, China), respectively. These probiotics had concentrations of 10^10^ CFU/g and were used in the fermentation process. The two commercially available enzymes (pectinase and cellulase) were courteously supplied by VTR Bio-Tech (Zhuhai, China). The cellulase and pectinase enzyme activities were 750,000 U/g and 50,000 U/g, respectively. The standard dose rates for cellulase and pectinase enzymes were 0.5 g and 0.3 g per kg, respectively. The commercial enzyme (xylanase) was bought from Sunson Biotech Group (Beijing, China). The xylanase enzyme activity was 30,000 U/g. The standard dose rate for xylanase enzyme was 0.3 g per kg. The enzymes were mixed with deionized water to make an enzyme solution before being used for hydrolysis. The enzymes were weighed accurately and mixed with 3 mL of deionized water with the help of an oscillator for about 10 min. After that, the enzyme solution was centrifuged for 5000× *g* for 10 min and the supernatant was used for the hydrolysis of BG in further experiments.

### 2.2. Brewer’s Grains Hydrolysis and Bacterial Fermentation

The dried brewer’s grain (DBG) was bought from the local brewery. Prior to the experiment, DBG was ground using a centrifugal grinder and screened with a 40-mesh screen for separation of big and tiny particles. The bigger particles were crushed again and separated utilizing the same mesh. Ultimately, the homogenized and tiny particles were chosen for further experimentation.

The fermentation of DBG was carried out in 150 mL conical flasks with butyl rubber corks and wrapped with aluminum foil. There were three treatments, namely, the control group, with DBG only (CON); DBG with enzymes only (BGE); and DBG with enzymes and *Lactobacilli* bacteria (BGEL).

QC01, QC02, and QC03 represent quality control samples. H37, H38, and H39 denote the CON group. H44, H45, and H46 represent the BGE group. H65, H66, and H67 belong to the BGEL treatment.

All the treatments had three replicates. Each flask contained 10 g of DBG and 37 mL of deionized water. A 3 mL sample of enzyme solution was added in both treatments, i.e., BGE and BGEL, at the start of the experiment, whereas 3 mL of deionized water was added in the CON treatment, and all flasks were placed onto a shaker running at 150 rpm at 50 °C for 28 h. After 28 h, 2 g of *Lactobacilli* mixture (1 g of *L. plantarum* and 1 g of *L. reuteri)* was added to the BGEL only and placed into a shaker again for an additional 48 h. At the end of the experiment, brewer’s grain (BG) samples were collected and stored at −80 °C for further analysis.

### 2.3. Metabolome Analysis

The metabolome analysis was carried out using the techniques described earlier [30,31,32]. Briefly, the 2 mL centrifugal tube was filled with 50 mg of BG sample, followed by a 6 mm crushing bead. Metabolite retrieval was performed using a 400 μL extraction solution (methanol: water = 4:1 *v*/*v*) containing 0.02 mg/mL of an internal standard (L-2-chlorophenylalanine). All samples underwent grinding for 6 min (−10 °C, 50 Hz) using a Wonbio-96c frozen tissue grinder (Shanghai Wanbo Biotechnology, Shanghai, China), followed by 30 min of low-temperature ultrasonic purification (5 °C, 40 kHz). The samples were kept at −20 °C for 30 min before being centrifuged for 15 min (4 °C, 13,000 g), and the supernatant was then transferred to an injection tube for high-performance liquid chromatography–mass spectrometry (HPLC-MS) analysis using a ThermoFisher HPLC system (Thermo Fisher Scientific Inc., Waltham, MA, USA).

The metabolites were identified by searching the HMDB database (http://www.hmdb.ca/ (accessed on 10 April 2024)) and the KEGG database (http://www.genome.jp/kegg (accessed on 10 April 2024)). The data matrix obtained by searching the database was uploaded to the Majorbio cloud platform (https://cloud.majorbio.com (accessed on 15 April 2024)) for data analysis. The R package “ropls” (Version 1.6.2) was used to perform principal component analysis (PCA) and orthogonal least partial squares discriminant analysis (OPLS-DA), with 7-cycle interactive validation used to evaluate the stability of the model. The metabolites with VIP > 1 and *p* < 0.05 were determined as significantly different metabolites based on the variable importance in the projection (VIP) obtained by the OPLS-DA model and the *p*-value generated by Student’s *t*-test. Differential metabolites among the two groups were mapped into their biochemical pathways through metabolic enrichment and pathway analysis based on the KEGG database (http://www.genome.jp/kegg/pathway (accessed on 23 April 2024)). The Python package “scipy.stats” (https://docs.scipy.org/doc/scipy/ (accessed on 20 May 2024)) was used to perform enrichment analysis to obtain the most relevant biological pathways for experimental treatments.

### 2.4. Metagenome Analysis

For metagenomic analysis, DNA extraction, metagenome sequencing, and functional annotations were performed according to the protocols described earlier [29]. A 1 mL BG sample was used for DNA extraction. Briefly, total genomic DNA was extracted from BG samples, according to the manufacturer’s instructions, using the Mag-Bind ^®^ Soil DNA Kit (Omega Bio-tek, Norcross, GA, USA). The DNA extract was fragmented for metagenome library construction using NEXTFLEX Rapid DNA-Seq (Bio Scientific, Austin, TX, USA). Then, sequencing was performed on Illumina NovaSeq 6000 (Illumina Inc., San Diego, CA, USA) according to the manufacturer’s instructions (www.illumina.com (accessed on 25 April 2024)).

The paired-end Illumina reads were trimmed of adaptors, and low-quality reads (length < 50 bp or with a quality value < 20 or having N bases) were removed by fastp [33] (https://github.com/OpenGene/fastp (accessed on 30 April 2024)). Metagenomics data were assembled using Megahit (v 1.1.2) [34]. Contigs with a length of ≥300 bp were selected as the final assembling result, and then the contigs were used for further gene prediction and annotation. Open reading frames (ORFs) from each assembled contig were predicted using Prodigal [35] (http://prodigal.ornl.gov/ (accessed on 10 May 2024)). A non-redundant gene catalog was constructed using CD-HIT [36] (http://www.bioinformatics.org/cd-hit/ (accessed on 15 May 2024)). High-quality reads were aligned to the non-redundant gene catalogs to calculate gene abundance with 95% identity using SOAPaligner [37] (http://soap.genomics.org.cn/ (accessed on 18 May 2024)). Representative sequences of the non-redundant (NR) gene catalog were aligned to the NR database using the Diamond [38] (http://www.diamondsearch.org/index.php (accessed on 20 May 2024)) for taxonomic annotations. The KEGG annotation was conducted using the Diamond [38] (http://www.diamondsearch.org/index.php (accessed on 20 May 2024)) against the Kyoto Encyclopedia of Genes and Genomes database (http://www.genome.jp/keeg/ (accessed on 20 May 2024)). The carbohydrate-active enzyme annotation was conducted using hmmscan (http://hmmer.janelia.org/search/hmmscan (accessed on 20 May 2024)) against the CAZy database (http://www.cazy.org/ (accessed on 20 May 2024)).

### 2.5. Analysis of Total Antioxidant Capacity, Flavonoids and Polyphenols

The total antioxidant capacity of BG was evaluated through the ABTS method (2,2′-azino-bis (3-ethylbenzothiazoline-6-sulfonic acid) and the DPPH method (2,2diphenyl-1-picrylhydrazyl), and the total flavonoid and polyphenol contents were determined using commercial kits following the manufacturers’ protocols [39,40]. These kits were bought from Suzhou Michy Biomed. Tech. Co., Ltd. (Suzhou, China)

The specimen’s antioxidant activity was quantified by measuring the variation in absorbance at 734 nm and using Trolox as a control agent. A 0.10 g BG sample was weighed, and 1 mL of extraction reagent was added. The homogenization of BG samples was carried out on ice, and then they were centrifuged for 10 min at 10,000 g and 4 °C. The liquid phase was obtained for additional analysis.

The variation in absorbance at 515 nm was determined, and Trolox was employed as a control solution to evaluate the antioxidant potential of the sample. A 0.10 g BG sample was weighed and 1 mL of extraction reagent was added. The samples were homogenized on ice and then centrifuged at 10,000× *g* and 4 °C for 10 min. The supernatant was collected for further examination.

For the calculation of the flavonoid content, the BG sample was air-dried to a uniform weight, crushed, and sieved over a 40-mesh screen. A 0.05 g BG sample was taken, and 1 mL of 60% ethanol was added. The mixture was mixed at 60 °C for 2 h before centrifugation at 10,000× *g* and 25 °C for 10 min. The supernatant was extracted for additional examination. The flavonoid content was calculated by measuring the absorbance of the sample extract at 510 nm.

For calculating the total polyphenol content, the BG sample was homogenized after drying to a consistent weight and screened through a 40-mesh screen. A 0.02 g BG sample was accurately weighed, and 1 mL of the extraction liquid was added. The solution was shaken for two hours at 60 °C, then centrifuged for ten minutes at 10,000× *g* and 25 °C. The supernatant was collected to conduct additional testing. The total phenol content of the sample was calculated by measuring the absorbance at 765 nm. The absorbances of all samples were measured using the microplate reader.

### 2.6. Statistical Analysis

The statistical assessments of microbial taxonomy, KEGG pathways, CAZymes, and metabolites were conducted utilizing the Mann–Whitney procedure, which has been integrated into the SPSS software version 20 (IBM Corporation, Armonk, NY, USA). The differences were defined as significant at *p* < 0.05, and following further correction of the acquired *p* values employing the false discovery rate (FDR), the significance level was established at an adjusted *p* value of ≤ 0.05. Utilizing the R software package (Version 1.6.2), the association between the impacted microbial species and metabolites was examined using the Spearman correlation test; only correlations with |r| > 0.6 and *p* < 0.05 were regarded as significant. The Gephi program was used to illustrate the correlation patterns (https://gephi.org/ (accessed on 25 May 2024)).

## 3. Results

### 3.1. Various Statistical Analyses of Samples

The samples’ correlation analysis revealed that the correlation coefficients of H65 with H66 and H67 were 0.9699 and 0.9778, respectively (Figure 1A). The BGEL had the highest number of total metabolites (1408), followed by CON (1369) and BGE (1362) (Figure 1B).

The PCA score analysis indicated that principal component (PC) 1 contributed to 41.00% of the variance, whereas PC 2 was responsible for 21.50% of the variance. Commutatively, PCs 1 and 2 were responsible for 62.50% of the variation in the data (Figure 1C). Furthermore, the PLS-DA analysis of the data revealed that Component 1 could account for 45.4% of the variance in the data, whereas Component 2 could predict 23.3% (Figure 1D). This indicated that there was a significant difference between CON and BGEL in metabolic composition, which was also evident from PCA analysis. The permutation test was used to assess PLS-DA’s modeling and prediction capabilities. The increasing trend of the regression line, together with declines in R2 and Q2, suggested that the model was not overfitted and was reliable. The R2 value of the PLS-DA plot was equal to 1, which is regarded as enormously substantial (Figure 1E).

### 3.2. Overview, Heatmap, and Correlation Analysis of Differentially Expressed Metabolites

We created volcano plots of the different metabolites in the different comparison groups, which revealed upregulated, downregulated, and non-significant metabolites. There were 35 upregulated and 69 downregulated metabolites observed in BGE compared to CON (Figure 2A). Comparing CON and BGEL, 99 upregulated and 113 downregulated metabolites were detected in BGEL (Figure 2B). However, comparing BGE and BGEL, the numbers of upregulated and downregulated metabolites were 112 and 83, respectively, in BGEL (Figure 2C). The Venn plot analysis of the differentially expressed metabolites (DEMs) in the multiple comparison groups indicated that each group exhibited distinguished metabolites, predicting changes in metabolic composition across the different groups. There were 35, 32, and 37 unique DEMs observed in CON vs. BGE, BGE vs. BGEL, and CON vs. BGEL, respectively. In comparison with CON, the BGEL exhibited the highest (212) number of DEMs, which suggested that *Lactobacillus* fermentation enhanced the expressions of different metabolites (Figure 2D).

An untargeted metabolomics analysis was performed employing LC-MS. This was intended to allow us to gain insight into the alterations that occurred throughout fermentation. A total of 511 differential metabolites were discovered, which could be classified as carbohydrates and their derivatives, TCA cycle intermediates, lipids and fatty acids, and amino acids and their derivatives.

The heatmap analysis revealed the upregulated metabolites in BGEL, which comprised amino acids and their derivates (Val-Gly-His, Val-Ser-Ile, ornithine, proline, and citrulline), organic acid (malic acid), fatty acids and lipids (butanoic acid, hydroxy-jasmonic acid, simonin, and prenylglucoside), and phenylpropanoids and polyketides (ononin, methyl flavanone, and benzopyran). The important downregulated metabolites in BGEL consisted of carbohydrates and carbohydrate conjugates (galactopyranuronosyl-L-rhamnose, glyceric Acid, D-erythrose 4-phosphate, D-digitoxose, and D-glucopyranoside) (Figure 3A).

The correlation analysis between metabolites revealed that L-proline had a positive correlation with butanoic acid and ethoxypropanoic acid, whereas it had a negative correlation with phenylalanyllysine and D-xylonic acid (Figure 3B). Similarly, hydroxyphenyllactic acid had a positive correlation with glucose propionate and hydroxybutyric acid, but had a negative correlation with hydroxyglutamate semialdehyde and methylmalonic acid. A positive correlation was also observed between L-sorbinose and 3-phenyllactic acid, and a negative correlation with ethynylestradiol was found.

### 3.3. KEGG and HMDB Compounds Classification

To make sense of the classification, the KEGG and HMDB databases were used for the classification of differentially expressed metabolites. A total of 120 metabolites were detected according to the KEGG classification. The KEGG database of phytochemical compounds indicated 37 chemicals, which were categorized as terpenoids, shikimate, polyketides, flavonoids, amino-acid-related compounds, phenylpropanoids, fatty-acid-related compounds, and others. Between them, alkaloids were the most diversified and abundant compounds (Figure 4A).

A total of 42 metabolites were detected according to the KEGG classification of compounds with biological roles. The carbohydrates were the most diversified and abundant compounds as compared to others (Figure 4B).

A total of 41 metabolites were detected according to the KEGG classification of lipid compounds. These compounds were categorized as fatty acyls, glycerophospholipids, polyketides, prenol lipids, sphingolipids, and sterol lipids. The fatty acyls were the most diversified and abundant compounds among them (Figure 4C).

According to the HMDB classification of compounds, 491 biochemicals were identified. The most abundant compounds were organic acids (31.57%), followed by organic oxygen compounds (16.90%). The phenylpropanoids and polyketides comprised 5.91%, which also included flavonoids and isoflavonoid compounds (Figure 4D).

### 3.4. KEGG Pathway Classification

The differential metabolites were mapped on the KEGG pathway database to identify the key metabolic pathways. These metabolic pathways were generally divided into metabolism, cellular processes, drug development, environmental information processing, human diseases, genetic information processing, and organismal systems. A total of 362 metabolic pathways were identified. The highest number of pathways was observed under the amino acid metabolism category, followed by the biosynthesis of other secondary metabolites and carbohydrate metabolism (Figure 5).

### 3.5. KEGG Enrichment Analysis and Differential Abundance Score

The differential metabolites were mapped on the KEGG database, and KEGG pathway enrichment analysis was performed to evaluate the biological roles of metabolites of different groups. The differential abundance score (DAS) was evaluated to assess the up- or downregulation of metabolites in a metabolic pathway. The enrichment ratio of the top 20 KEGG pathways was evaluated.

In CON vs. BGE, the highest enrichment ratio (*p* < 0.001) was observed in BGE for arginine and proline metabolism, followed by D-amino acid metabolism and mineral absorption (*p* < 0.01). The DAS of CON vs. BGE revealed that only phenylalanine metabolism-related metabolites were downregulated in BGE, whereas the pathways with upregulated metabolites were related to arginine and proline metabolism, protein digestion and absorption, galactose metabolism, mineral absorption, and D-amino acid metabolism (Figure 6D).

The KEGG enrichment analysis of CON vs. BGEL revealed that the metabolic pathways in BGEL were related to the galactose metabolism, and glycerolipid metabolism had the most prominent enrichment ratio (*p* < 0.001). The other key pathways with significantly higher (*p* < 0.05) enrichment ratios in BGEL were phenylalanine metabolism, biosynthesis of various plant secondary metabolites, biosynthesis of phenylpropanoids, arginine biosynthesis, isoflavonoid biosynthesis, carbohydrate digestion and absorption, and arginine and proline metabolism (Figure 6B). According to DAS analysis of CON vs. BGEL, the pathways exhibiting upward trends of metabolites in BGEL were related to arginine biosynthesis, isoflavonoid biosynthesis, carbohydrate digestion and absorption, and galactose metabolism (Figure 6E).

The KEGG enrichment analysis of BGE vs. BGEL revealed that metabolic pathways in BGEL related to the galactose metabolism and biosynthesis of phenylpropanoids had the highest enrichment ratios, which was significant (*p* < 0.001). The other key pathways with higher (*p* < 0.05) enrichment ratios in BGEL were isoflavonoid biosynthesis and biosynthesis of various other secondary metabolites (Figure 6C). The DAS of BGE vs. BGEL revealed that pathways with upregulated metabolites in BGEL were related to arginine biosynthesis, galactose metabolism, fructose and mannose metabolism, and isoflavonoid biosynthesis (Figure 6F). These results suggest that the *Lactobacillus* fermentation resulted in a high enrichment ratio of isoflavonoid and phenylpropanoid biosynthesis pathways in BGEL, which, in turn, elevated the antioxidant capacity of BGEL.

### 3.6. Heatmap and VIP Analysis of Metabolites Related to Antioxidant Activity

The metabolites exhibiting antioxidant-related activity (flavonoids, isoflavonoids, cinnamic acids, and prenol lipids) were screened and subjected to the OPLS-DA model, and the VIP score was evaluated to identify the significant variation in metabolites of the two groups. The expression profiles of CON and BGE revealed that six metabolites were identified and only methylflavanone was significantly higher (*p* < 0.01) in BGE (Figure 7A). There were a total of 15 metabolites identified in the expression profiles of BGE and BGEL. The daidzein, hydroxypiperitone, and ononin compounds were significantly elevated (*p* < 0.01) in the BGEL group. The highly significant metabolites (*p* < 0.001) in BGEL as compared to BGE were kaempferol, aucubin, eucarvone, isoartocarpesin, tectorigenin, dihydroxyflavone, diferuloylputrescine, methylflavanone, inuline, benzopyran, and indoleacetic acid (Figure 7B).

The expression profile of CON and BGEL demonstrated that the most prominent metabolites (*p* < 0.001) in BGEL were hydroxypiperitone, dihydroxyflavone, eucarvone, isoartocarpesin, aucubin, ononin, methylflavanone, benzopyran, indoleacetic acid, inuline, and kaempferol. Similarly, the significantly higher (*p* < 0.01) metabolites detected in BGEL as compared to CON were daidzein and diferuloylputrescine. The tectorigenin was also significantly different in BGEL as compared to CON (*p* < 0.05) (Figure 7C). These results suggested that *Lactobacillus* fermentation significantly enhanced the flavonoids, isoflavonoids, cinnamic acids, and prenol lipids in BGEL as compared to other groups, which led to an increase in its overall antioxidant capacity.

The proposed pathway of flavonoids and isoflavonoids is discussed in Figure 7D. The p-coumaric acid is formed from tyrosine or cinnamic acid in the phenylpropanoid biosynthesis pathway, which then enters the flavonoid biosynthesis pathway and is converted to isosakuranetin, kaempferol, and liquiritigenin. Later, isosakuranetin is transformed to 5,7-Dihydroxy-4′-methoxy-8-methylflavanone, kaempferol is transformed to kaempferol 3-neohesperidoside-7-(2″-p-coumarylglucoside) and kaempferol 3-neohesperidoside-7-(2″-p-coumaryllaminaribioside), and liquiritigenin is transformed to 4′,6-Dihydroxyflavone. After liquiritigenin enters the isoflavonoid biosynthesis pathway, it is converted to daidzein, tectorigenin, and ononin.

### 3.7. Correlation Analysis of Bacterial Community with Metabolites and CAZy

Spearman correlation was performed to check the association between microbial species and metabolites of BGEL. Interestingly, most of the species had positive associations with metabolites and belonged to the *Lactiplantibacillus* and *Aspergillus* genera.

Kaempferol, aucubin, and benzopyran-4-one were positively correlated with the *Lactobacillus* species; however, aucubin had a significant (*p* < 0.01), positive correlation (>0.99) with *Lactiplantibacillus plantarum*. Isoartocarpesin, ononin, tectorigenin, and methylflavanone were positively associated with *Aspergillus* species and *Limosilactobacillus reuteri;* however, isoartocarpesin possessed a significant (*p* < 0.01) positive correlation (1) with *Aspergillus parasiticus*. The *Saccharomyces cerevisiae* and *Kroppenstedtia eburnea* had a positive correlation with diferuloylputrescine, 4′,6-dihydroxyflavone, and inuline (Figure 8A). The above results showed that *Lactobacillus spp.* had a direct positive association with flavonoids and isoflavonoids, which led to an increased antioxidant capacity of BGEL.

Spearman correlation network analysis was performed to check the association between microbial species and CAZy of BGEL. Interestingly, most of the species had an association with CAZy and belonged to the *Lactiplantibacillus* and *Aspergillus* genera. A total of four classes of enzymes exhibited correlations with microbial species of BGEL, namely, glycoside hydrolases (GH), glycosyl transferases (GT), carbohydrate esterases (CE), and auxiliary activities (AA) (Figure 8B).

*Lactiplantibacillus plantarum* had significant (*p* < 0.05) and positive correlations with GH25, GT5, and CE9, whereas *Limosilactobacillus reuteri* had a significant (*p* < 0.05) negative correlation with GH32. *Lacticaseibacillus paracasei* possessed significant (*p* < 0.05) and positive correlations with CE1, GH36, AA6, GT26, and GH13. *Aspergillus oryzae* exhibited significant (*p* < 0.05) and positive correlations with AA7, GH42, CE2, AA3_2, GH18, GH76, and GH92. The larger node in the network analysis denotes the higher abundance of the species and its function contribution. g__Lactiplantibacillus had the largest node, followed by the g__Lacticaseibacillus, g__Limosilactobacillus, and g__Aspergillus, indicating that these genera and their related species may be responsible for the enzyme production during lactobacillus fermentation of BGEL.

### 3.8. Antioxidant Capacity and Content of Flavonoids and Polyphenols 

The total antioxidant capacity of fermented BG was determined by a DPPH and ABTS assay. The DPPH and ABTS scavenging percentages of CON and BGE were comparable. However, the DPPH and ABTS scavenging percentages of BGEL were significantly higher (*p* < 0.05) as compared to CON and BGE (Figure 9A, B). The total flavonoid and polyphenol contents of BGEL were remarkably greater (*p* < 0.05) than those of the other groups (Figure 9C and Figure 9D). There was no variation in the total flavonoid or polyphenol contents of CON and BGE. The higher contents of flavonoids and polyphenols in the BGEL might have contributed to its increased overall antioxidant capacity.

## 4. Discussion

The integration of metabolomics with metagenomics has become an effective tool for understanding complex biological processes and discovering new biomarkers that have important implications in many disciplines. Our work utilized this integrated strategy to examine the antioxidant capacity of fermented brewer’s grains. These products possess good antioxidant capacity and have the potential to be used as a functional food component and as a bioactive ingredient in animals’ diets to reduce toxicity due to their better nutritional profiles and bioactive chemicals [41,42].

*Lactobacillus* is a Gram-positive bacterium that can thrive in both anaerobic and aerobic environments. It can ferment carbohydrates to produce two different isomers of lactic acid [23]. It has been utilized as a probiotic frequently because of its remarkable probiotic, antioxidant, and immune-enhancing capabilities [24,43]. Our research work, utilizing integrated metabolomic and metagenomic analysis, provides the first insights into the specific features of the metabolome and metagenome, as well as the antioxidant-related metabolites, in brewer’s grains that have been inoculated with *L. plantarum* and *L. reuteri*.

### 4.1. Effect on Metabolome of Fermented Brewer’s Grains

Our metabolomics analysis provides comprehensive insights into the metabolic profiles of fermented brewer’s grains. Through high-throughput techniques, i.e., high-performance liquid chromatography–mass spectrometry (HPLC-MS), we identified a diverse array of metabolites, including phenolic compounds, flavonoids, isoflavonoids, cinnamic acids, prenol lipids, organic acids, and amino acids. These metabolites are known for their antioxidant properties and play crucial roles in scavenging free radicals, reducing oxidative stress, and conferring health benefits [44,45].

The heatmap indicated a significant upregulation of isoflavonoids and flavonoids in the BGEL group. Notable flavonoids, such as dihydroxyflavone, methylflavanone, benzopyran, isoartocarpesin, and kaempferol derivatives, were significantly upregulated in BGEL. The flavonoids have been well-documented for their role in reducing the risk of chronic diseases through their antioxidant and anti-inflammatory properties [46,47,48]. Dihydroxyflavone (flavones) demonstrates antioxidant activity, as highlighted by Chen et al. [49], who elucidated its neuroprotective effects against glutamate-induced toxicity through antioxidant properties. The antioxidant activity of methylflavanone (O-methylated flavonoids) may involve mechanisms such as direct scavenging of free radicals and induction of cellular detoxification enzymes, as suggested by Tsuji et al. [50] in their structure–activity analysis of flavonoids. Benzopyrans (O-methylated flavonoids) possess antioxidative activities, which may contribute to their potential therapeutic applications in oxidative stress-related conditions [51]. The isoartocarpesin is a flavone that acts as an antioxidant by mitigating excessive reactive oxygen species (ROS) production [52]. The kaempferol derivatives are flavonoid glycosides that demonstrate diverse bioactivities, including antioxidant, antitumor, and anti-inflammatory activities [53]. This suggests that *Lactobacillus* fermentation enhances the content of flavonoids and, hence, the antioxidant quality of BGEL.

Isoflavonoids like tectorigenin, daidzein, and ononin showed significant increases in their expression in BGEL compared to CON and BGE. These isoflavonoids are known for their antioxidant properties, which contribute to reducing oxidative stress [54,55,56]. Tectorigenin is a natural compound, classified as an isoflavone, which possesses various biological activities, including antioxidant, anti-inflammatory, anti-cancer, and estrogenic effects [57]. Daidzein is an isoflavone that exerts its antioxidant mechanism by inhibiting lipid peroxidation in cellular membranes against nephrotoxicity [58]. Ononin is a naturally occurring isoflavone glycoside generally found in various plants. It exhibits antioxidant properties, which are attributed to reducing ROS generation and the levels of pro-inflammatory markers [59]. The increased levels of these isoflavonoids suggested that *Lactobacillus* fermentation enhanced the antioxidant capacity of BGEL.

Indoleacetic acid, aucubin, eucarvone, hydroxypiperitone, and inuline belong to the class of prenol lipids. Importantly, indoleacetic acid (IAA) (monoterpenoids) has been found to be synthesized by microbiota from dietary tryptophan [60]. The IAA revealed anti-inflammatory and antioxidant activity as a result of induction of HO-1 and direct neutralization of free radicals [61]. Aucubin, an iridoid glycoside (terpene glycoside), is recognized for its hepatoprotective, anti-inflammatory, and antioxidant properties [62]. The eucarvone also belongs to the group of monoterpenoids and exhibits antioxidant activity, as evidenced by its ability to scavenge free radicals and reduce oxidative stress in biological systems [63,64]. Hydroxypiperitone (monoterpenoids) exhibits antioxidant properties due to its ability to scavenge free radicals, thus potentially mitigating oxidative stress-related damage, as evidenced by previous studies [65,66,67]. Inuline is a class of diterpenoid alkaloids which act as antioxidants by scavenging free radicals and enhancing antioxidant enzyme activity [68]. The fermentation process also resulted in the upregulation of these prenol compounds, which may have enhanced the antioxidant properties of BGEL.

Various researchers have used *Lactobacillus* fermentation to enhance the nutritional value of and bioactive substances in fermented brewer’s grains. Similar results have been achieved in our study. Pejin et al. [69] investigated lactic acid fermentation of brewer’s spent grain by *Lactobacillus rhamnosus* to utilize reducing sugars for the production of lactate. The *Lactobacillus* fermentation of brewer’s grains significantly alters the metabolites, as evidenced by the increased propionate concentration observed in *Lactobacillus* mucosae-fermented wet brewers’ grains [70]. Moreover, in another experiment, the LAB fermentation of BSG with *Lactiplantibacillus plantarum* increased the phenolic compounds in fermented BSG [71]. These findings underscore the potential of *Lactobacillus* fermentation to enhance the nutritional quality and utilization of brewer’s grains.

The fermentation of brewer’s grains by Lactobacillus impacts several metabolic pathways, as shown by KEGG enrichment analysis. Research has shown that the process of *Lactobacillus* fermentation improves the development of pathways related to the breakdown of carbohydrates, such as glycolysis and the pentose phosphate pathway [72]. In addition, the fermentation of brewer’s grains by *Lactobacillus* influences the pathways involved in the metabolism of amino acids, including their synthesis and breakdown, which aligns with our experiment. This might potentially impact the nutritional value of the fermented product [73].

*Lactobacillus* fermentation of brewer’s grains impacts pathways involved in the creation of bioactive chemicals and vitamins. *Lactobacillus* fermentation improves the enrichment of pathways involved in the manufacturing of B vitamins, including riboflavin and folate, which are essential for a variety of cellular processes [74]. Furthermore, bioactive substances produced during *Lactobacillus* fermentation, such as antioxidants and antimicrobial peptides, might affect pathways linked with oxidative stress responses and immunological modulation, possibly improving the fermented product’s functional qualities, as observed in the current experiment [75,76].

The influence of *Lactobacillus* fermentation on the isoflavonoid biosynthetic pathway in brewer’s grains has been widely explored. Our experiment revealed that the biosynthesis pathways isoflavonoids and phenylpropanoids show high enrichment ratios in BGEL, leading to increased antioxidant capacity. Similar results have been reported previously. Research by Merenkova et al. [10] emphasized the formation of isoflavone aglycones in fermented cereal substrates, demonstrating the possibility to improve isoflavonoid concentration through fermentation methods. Similarly, Verni et al. [77] explored the alteration of the biosynthesis pathways of flavonoids during fermentation, showing a rise in isoflavone concentration post-fermentation. Yang et al. [78] further supported these results, demonstrating a considerable increase in isoflavone levels in polyphenol-rich foods during fermentation.

Several researchers have studied the influence of fermentation processes, notably with lactic acid bacteria, on the alteration of flavonoids and isoflavonoids in different food products. In one study, Gao et al. [79] studied flavonoid biosynthesis during alfalfa silage fermentation and found an increase in antioxidant activity post-fermentation. These results together reveal the delicate interaction between fermentation, isoflavonoid production, and antioxidant characteristics, clarifying the potential of fermentation techniques to increase the nutritional content and bioactivity of food items.

The phenylpropanoid production pathway plays a vital role in boosting antioxidant properties as a result of *Lactobacillus* fermentation, as indicated by various researchers. Meng et al. [80] revealed that the combined impact of hydrolysis of proteins and Lactobacillus fermentation substantially improves the antioxidant capacity and metabolomic characteristics of quinoa beverage, suggesting modifications in phenylpropanoid concentration. Similarly, Lei et al. [81] studied the impact of *Latilactobacillus delbrueckii* fermentation on phenolic substances’ biological conversion and antioxidant activities in quinoa seeds, indicating higher antioxidant activity linked to alterations in the phenylpropanoid biosynthesis process. The alteration of phenylpropanoid concentration during *Lactobacillus* fermentation occurs via enzymatic processes of *Lactobacillus* spp. This process comprises enzymatic changes, such as hydroxylation, methylation, and glycosylation, resulting in the creation of various phenylpropanoid derivatives with increased antioxidant capabilities. Additionally, fermentation environments, including pH, temperature, and substrate availability, may impact the functioning of genes associated with phenylpropanoid biosynthesis, further enhancing the phenolic profile of the final product [82,83,84]. These results together underline the relevance of the phenylpropanoid production pathway in boosting antioxidant potential during *Lactobacillus* fermentation, which is in line with the current experiment.

### 4.2. Correlation Analysis of Bacterial Community with Metabolites and CAZy

By integrating metabolomics and metagenomics data, we elucidated the intricate interplay between microbial metabolism and the production of bioactive metabolites in fermented brewer’s grains. Our analysis identified specific metabolite–microbe associations, highlighting key metabolic pathways and enzymatic activities responsible for the biosynthesis of antioxidants. Furthermore, Spearman correlation analyses unveiled potential synergistic interactions among microbial taxa and metabolites, providing mechanistic insights into the antioxidant capacity of the fermentation process.

During fermentation, *L. plantarum* can promote the liberation and conversion of flavonoids by enzymatic reactions that degrade complicated polyphenols into simpler ones that can be absorbed more easily by the body [85]. Previous investigations have demonstrated that the process of *L. plantarum* has a positive association with flavonoids like daidzein, kaempferol, genistin and vanillic acid, and LAB fermentation improved the overall flavonoid percentage and modulated the composition of individual flavonoids, which led to enhanced antioxidant capacity and tyrosinase inhibition [86]. The *Lactobacillus plantarum* fermentation also exhibited a positive relationship with phenolic characteristics of the kiwifruit pulp by increasing the quantity of dihydroxy coumarin and, specifically, protocatechuic acid, which enhanced the scavenging and antioxidant capability of fermented kiwifruit [87]. These modifications in phenolic substances might be attributed to the glucosidase enzyme released by *L. plantarum*, causing changes in the phenolic structure [88]. Feruloyl esterase enzymes accelerate the release of ferulic or p-coumarate from bound phenols. *L. plantarum* releases the phenolic acid decarboxylase enzyme, which transforms phenolic molecules and influences the overall antioxidant capacity [89].

Additionally, our correlation analysis has demonstrated strong relationships between the prevalence of LAB and increased concentrations of particular flavonoids, revealing a direct association between microbial growth and the enhancement of flavonoids.

*Lactobacillus* fermentation also affects the carbohydrate-active enzyme (CAZy) profile, promoting the digestion of complex polysaccharides and enhancing the accessibility and availability of nutrients [90]. The species and CAZy correlation analysis offered detailed information on the vast variety of enzymes employed by *Lactobacillus* fermentation, such as glycoside hydrolases, glycosyl transferases, carbohydrate esterases, and auxiliary activities. These enzymes are necessary for the decomposition of complex carbohydrates contained in the substrate, such as plant cell wall polysaccharides, into fermentable sugars like glucose and galactose. This enzymatic activity eventually leads to the synthesis of numerous fermentation products, including organic acids, flavor and bioactive compounds, impacting the sensory characteristics and nutritional value of fermented foods [91,92,93]. Recent investigations have indicated that some CAZy families have significant connections to the metabolic processes of *Lactobacillus* species, affecting their capability to degrade complicated carbohydrates into less complex sugars [94]. The existence of glycoside hydrolases has been associated with the greater abundance of *Lactobacillus plantarum* in the fermentation process, demonstrating a synergistic association that promotes the efficacy of fermentation [95]. Numerous CAZy were found to have a high association with microbial species in the current investigation, including glycoside hydrolases, glycosyl transferases, and carbohydrate esterases, which may have contributed to the enhanced nutritional value of BGEL.

### 4.3. Effect on Antioxidant Capacity and Flavonoid and Polyphenol Content

Lactic acid bacteria, namely, *Lactobacillus spp*., generate metabolites such as exopolysaccharides, phenolic substances, and organic acid compounds. These compounds are the foundation for the fermented product’s antioxidant characteristics [96]. The results of our investigation show that these chemicals, which possess antioxidant capabilities, significantly increase the fermented product’s antioxidant capacity by efficiently neutralizing damaging free radicals and preventing oxidative stress.

Flavonoids and polyphenols are biologically active substances well-known for their antioxidant and health-promoting potential [97]. Research has revealed that the process of *Lactobacillus* fermentation might result in a significant elevation of the flavonoid and polyphenol levels in fermented products [98]. Meng et al. [85] conducted research in which they demonstrated that the process of fermentation using *Lactobacillus* led to a substantial rise in the flavonoid profile of loquat juice, which eventually led to elevated antioxidant capability. Verni et al. [99] observed a substantial increase in the polyphenol and flavonoid concentration and antioxidant capability of brewer’s grain after it had been fermented with *Lactobacillus plantarum*.

The scavenging percentage of BGEL was increased after *Lactobacillus* fermentation, which is in line with the results reported previously [99]. *Lactobacillus* cultures exhibit enzymatic properties that can break down complicated substances found in brewer’s grain into simpler molecules, thus increasing the accessibility of bioactive molecules [77,100]. Furthermore, the process of *Lactobacillus* fermentation may result in the creation and release of antioxidant molecules via metabolic processes, including the transformation of phenolic precursors into bioactive compounds [101]. In addition, the fermentation process creates acidic conditions that improve the extraction and availability of bioactive molecules, thus raising their concentrations in the fermented product [77]. These might be reasons for the increasing scavenging property of BGEL.

## 5. Conclusions

*Lactobacillus* fermentation of brewer’s grains substantially modifies their metabolic profile, increasing the concentrations of various bioactive compounds. This enhancement in metabolites such as isoflavonoids, flavonoids, and phenolic acids suggests that fermented brewer’s grains could offer superior health benefits compared to unfermented grains. These findings highlight the potential of using fermented brewer’s grains in animal feed and functional foods to leverage their improved nutritional and health-promoting properties. By boosting the nutritional value of BSG, fermented BSG could potentially find applications in the food industry, serving as a culture medium, a functional food components for human consumption, and a bioactive feed ingredient for animals. Future studies should focus on the specific mechanisms through which *Lactobacillus* fermentation enhances these metabolites and the subsequent health effects observed in vivo and in vitro.

## Figures and Tables

**Figure 1 antioxidants-13-00872-f001:**
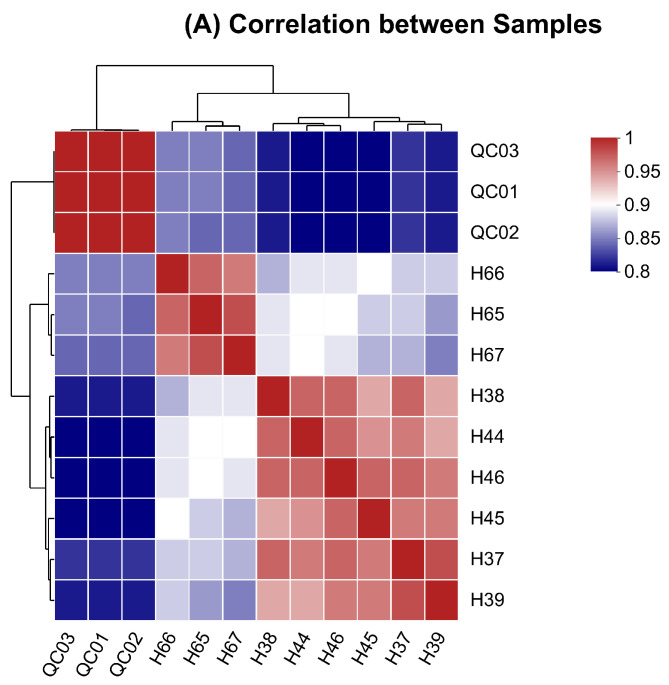
Sample analysis: (**A**) correlation analysis between samples; (**B**) Venn diagrams of different groups; (**C**) PCA scores analysis; (**D**) PLS-DA scores plot; and (**E**) permutation test of PLS-DA. The individual grid in (**A**) depicts the correlation between the two samples. The different colors in (**A**) show the values of the correlation coefficient among the samples, which varied from 0.8 to 1. The different colors of Venn diagrams represent different groups.

**Figure 2 antioxidants-13-00872-f002:**
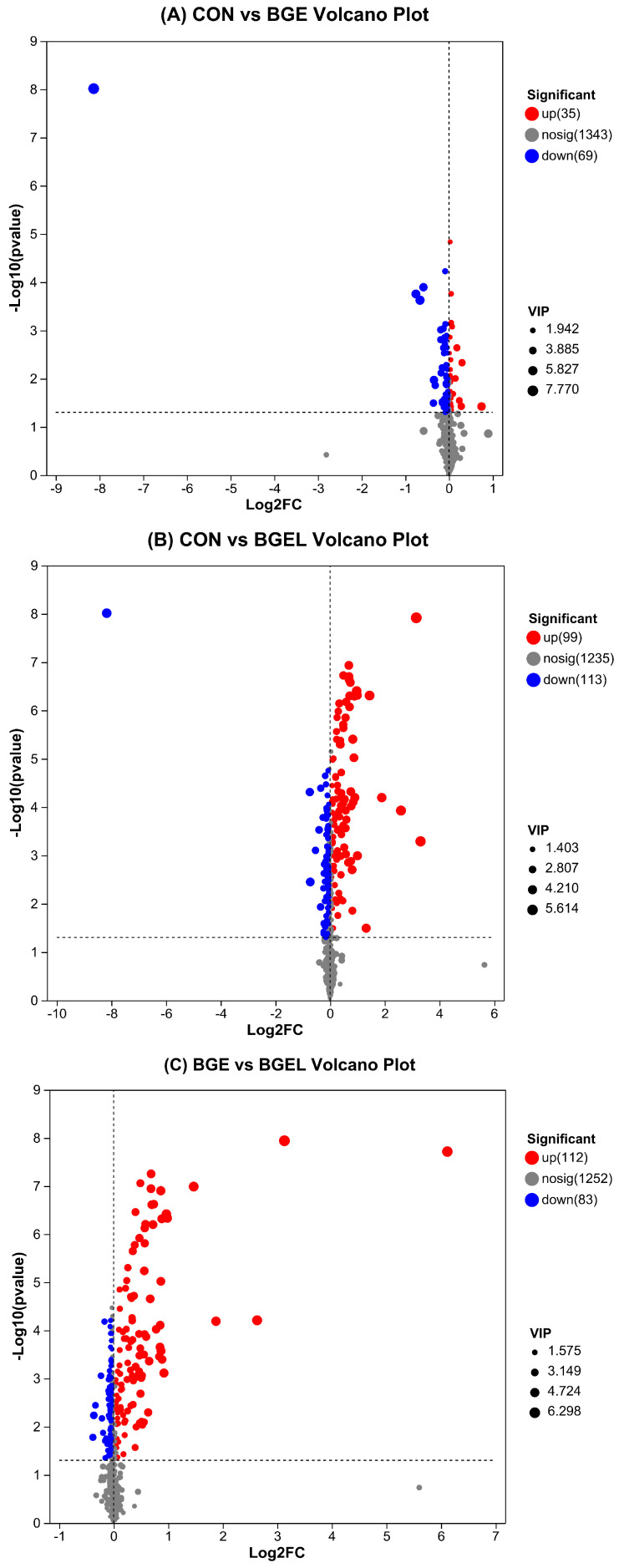
Overview of differentially expressed metabolites: (**A**) volcano plot between CON and BGE; (**B**) volcano plot between CON and BGEL; (**C**) volcano plot between BGE and BGEL; and (**D**) Venn diagram analysis.

**Figure 3 antioxidants-13-00872-f003:**
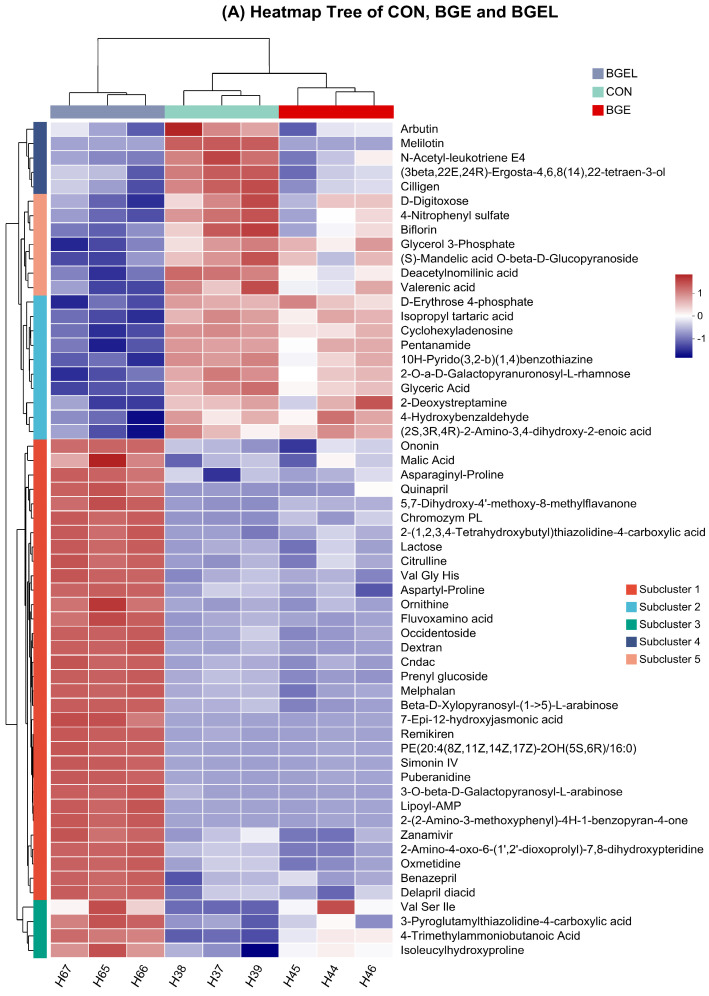
Heatmap tree and correlation analysis: (**A**) heatmap analysis of CON, BGE, and BGEL and (**B**) correlation analysis of metabolites.

**Figure 4 antioxidants-13-00872-f004:**
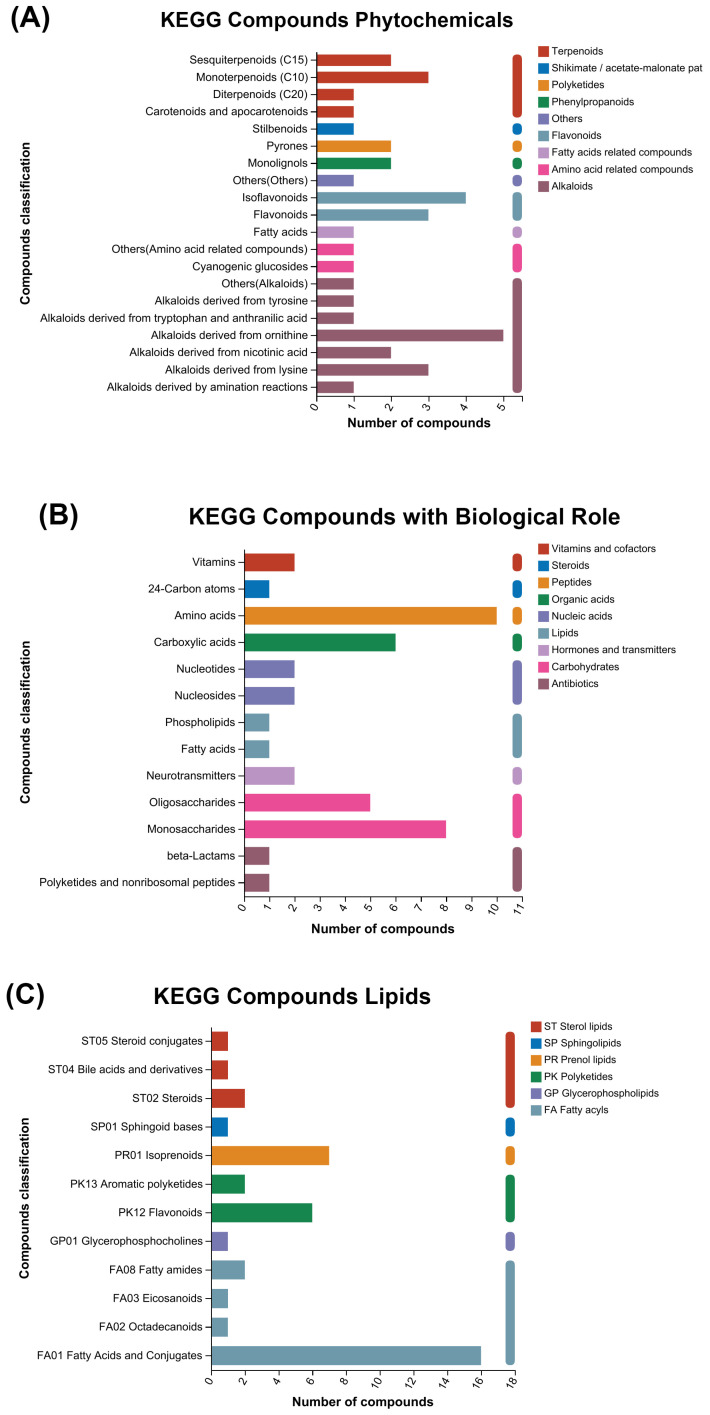
Classification of KEGG and HMDB compounds: (**A**) classification of phytochemical compounds; (**B**) classification of compounds with biological roles; (**C**) classification of lipids; and (**D**) classification of HMDB compounds.

**Figure 5 antioxidants-13-00872-f005:**
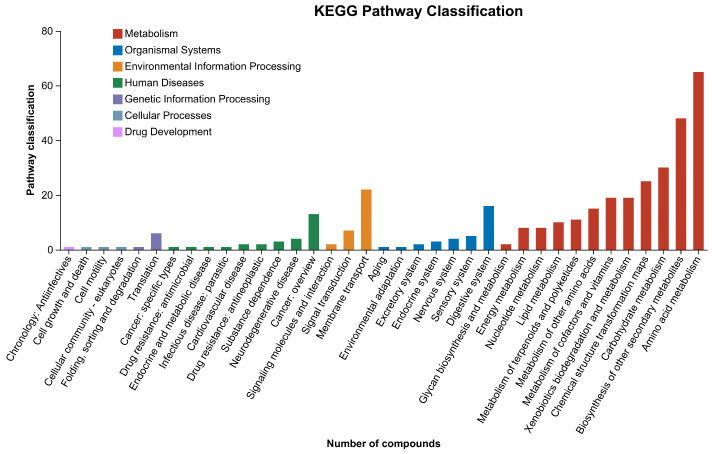
Overall KEGG pathway classification.

**Figure 6 antioxidants-13-00872-f006:**
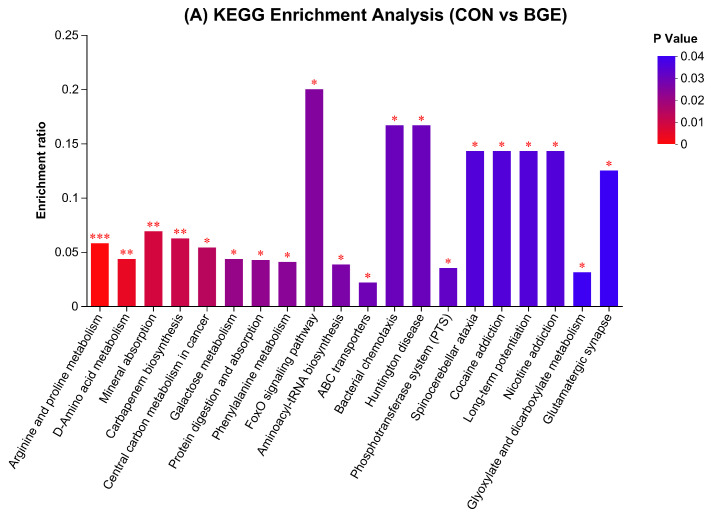
KEGG enrichment analysis and differential abundance score: (**A**) KEGG enrichment analysis (CON vs. BGE); (**B**) KEGG enrichment analysis (CON vs. BGEL); (**C**) KEGG enrichment analysis (BGE vs. BGEL); (**D**) differential abundance score (CON vs. BGE); (**E**) differential abundance score (CON vs. BGEL); and (**F**) differential abundance score (BGE vs. BGEL). *** *p* < 0.001; ** *p* < 0.01; * *p* < 0.05.

**Figure 7 antioxidants-13-00872-f007:**
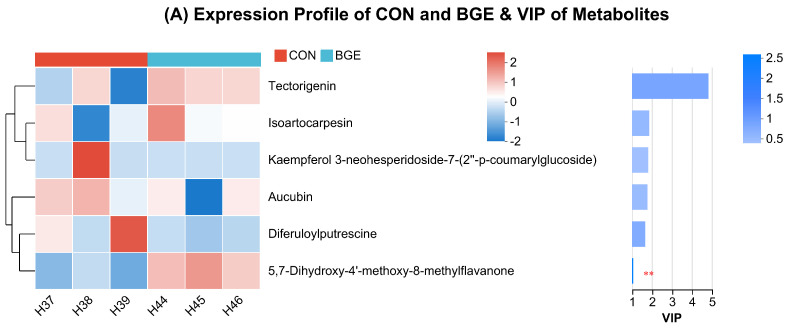
Analysis of metabolites related to antioxidant activity: (**A**) heatmap and VIP analysis of CON vs. BGE; (**B**) heatmap and VIP analysis of BGE vs. BGEL; (**C**) heatmap and VIP analysis of CON vs. BGEL; and (**D**) proposed biosynthesis pathway of flavonoids and isoflavonoids. The red boxes indicate the upregulated metabolites. *** *p* < 0.001; ** *p* < 0.01; * *p* < 0.05.

**Figure 8 antioxidants-13-00872-f008:**
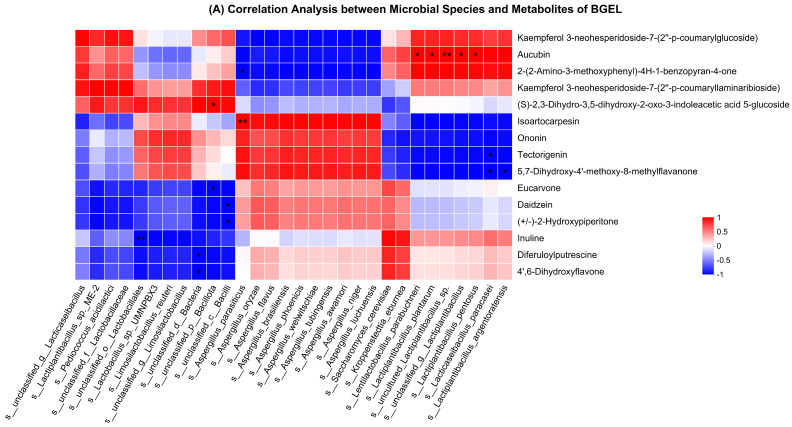
Spearman correlation analysis. (**A**) Correlation analysis between microbial species and metabolites of BGEL and (**B**) correlation analysis between microbial species and CAZy of BGEL. ** *p* < 0.01; * *p* < 0.05.

**Figure 9 antioxidants-13-00872-f009:**
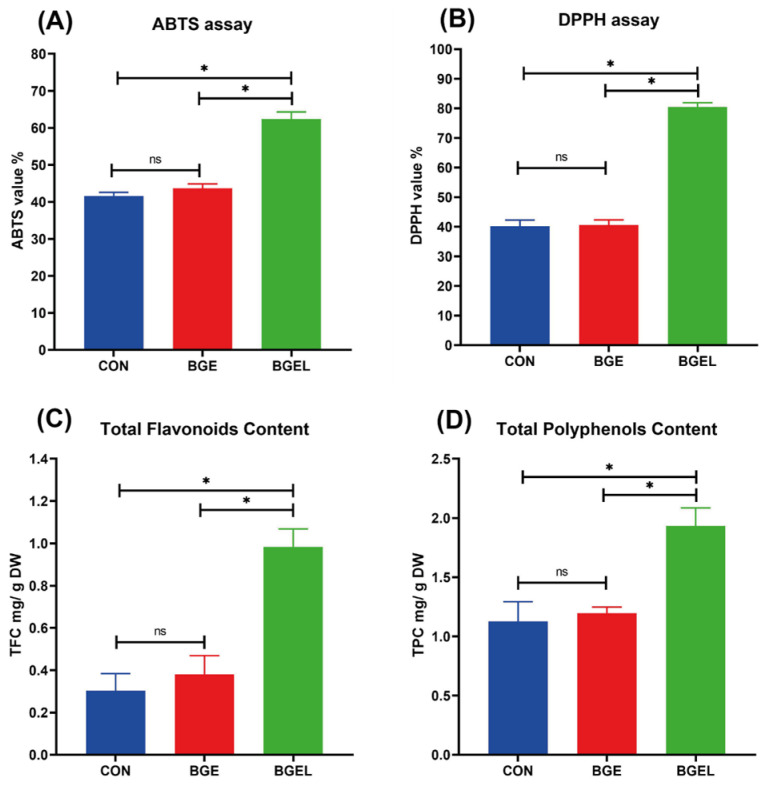
Scavenging values, total flavonoids, and total polyphenols: (**A**) ABTS assay to evaluate the scavenging percentage; (**B**) DPPH assay to evaluate the scavenging percentage; (**C**) total flavonoid content; and (**D**) total polyphenol content. * *p* < 0.05; ns non-significant.

## Data Availability

Data will be available upon request.

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
