# Peer review of "Integrated Metabolomics and Metagenomics Unveiled Biomarkers of Antioxidant Potential in Fermented Brewer’s Grains"

_antioxidants, 2024, doi:10.3390/antiox13070872_

Round 1

Reviewer 1 Report

The article has great potential and focuses on one of today's global problems. The valorization of food industry waste. However, the authors have to make some changes to bring out the full potential of the article. First of all, in the introduction the part of metagenomics is not perfectly indicated. All the figures need to be improved, they have a low quality and the name labels do not look good due to the small size. In the results it is difficult to interpret the differences between the groups. I recommend the authors to be more concise, shorten the text of the results and highlight the most important, making clear the differences between groups and if the use of lactobacillus is really a benefit. This is the most important part of the work and its benefit is not clear. The discussion is also too long, the authors should be more concise, with half of the pages of discussion would be enough. Also the number of references is too high for a research article.

Line 41: A reference for the food waste?

Line 53: The authors have to add a reference for “rich in nutrients”.

Line 60-61: I don´t understand the reference to microorganisms here.

Line 79: Prevent the production of toxins? I guess the authors want to say that the use of bacteria can lead to the present of toxins in the final product.

Line 116: Why these strains were used? In what was based the selection?

Line 123: Why the authors used that concentration of enzymes? Is based in previous studies? This is an important part of the work.

Line 128: Why was centrifuged?

Line 144: A mixture. O.5 g of each strain?

Line 160: If other databases were used should be also indicated.

Line 163: And the version of R?

Line 177: Amount of sample used for DNA isolation?

Line 180: Delete here Novaseq to avoid repetition.

Line 206: I don´t understand this sentence here. The same comment for line 213. The important is what the authors used.

Line 231: Please, indicate the equipment used to measure absorbance.

Line 232: I don´t understand why the authors used two statistical programs when with R the authors could do all the analysis.

Figure 1: The quality of images is low and due to their small size, it is impossible to interpret the results. Please modify this.

Line 244: The names of the samples should be indicated in M&M section. Also, the name used for sample is not intuitive

Line 249: Delete this sentence. It is not necessary to explain this. Also the explanation of the color should be in the Figure title, not in the main text.

Line 252.254: The same comment as before. That information should ben indicated in the figure, not in the main text. Please, focus here only in the important results.

Line 258-267: Ok, but what variables are the main responsible of the difference between samples.

Maybe section 3.2 and 3.3 could be joined in one section as both are focused in metabolites.

Line 302-307: Upregulated in what group and downregulated in what group? Please clarify.

Figure 3, Figure 4, Figure 6: It is impossible to interpret the results because the labels with the names are too small.

Line 432: Lactobacillus in italics.

Line 452: The authors should add some information of the metagenomic results. I mean the authors should compare the communities of the three groups, to have a better interpretation of the correlation indicated in that section. Also, it is too difficult to interpret the results of this section. Too much text. I think the authors should be more concise.

Line 803: Lactobacillus in italics.

Author Response

Dear Reviewer,

Regards,

Reviewer 2 Report

The paper “Integrated Metabolomics and Metagenomics Unveiled Biomarkers of Antioxidant Potential in Fermented Brewer’s Grains” is focused on the metabolic alterations and processes in the fermentation of Brewer's Spent Grain by microbes. The authors hypothesize that breaking down fibers with enzymes could be useful to boost lactobacilli fermentation and increase the antioxidant and nutritional potential of BSG. Although the work is well-written and structured, and the methods are clearly described, the figures present a low-resolution and illegible text. 

The authors are advised to improve the resolution of all pictures to make data readable.

Author Response

Dear Reviewer,

Regards,

Reviewer 3 Report

In this manuscript, using a combination of modern analytic tools (metabolomics and metagenomics), the authors assessed the impact of Lactobacilli fermentation on brewer's grains metabolic profile and pathways. More specifically, they discovered that Lactobacillus fermentation significantly boosted the antioxidant capabilities as well as the nutritional value of brewer's grains. This is very interesting as using the process described here a "less valuable" waste could potentially be turned into highly valuable and protective ingredients. I would recommend acceptance after the following points are addressed by the authors.

1. The quality of the figures is very poor. Please provide Figures with higher quality.

2. In the fermentation/enzymatic hydrolysis experiments, the authors used DI water as medium. Given that enzymes are highly pH-sensitive, how did the authors ensure that there is no loss of activity during the process. Why not use a buffer in place of DI water.

3. Fig. 9 shows that in the absence of fermentation, the antioxidant activity of BGE sample is comparable to that of the control sample (CON) even though well-known biodegradation enzymes such as cellulase are present. Is this outcome due to lack of degradation or to formation degradation products with no antioxidant activity?

In this manuscript, using a combination of modern analytic tools (metabolomics and metagenomics), the authors assessed the impact of Lactobacilli fermentation on brewer's grains metabolic profile and pathways. More specifically, they discovered that Lactobacillus fermentation significantly boosted the antioxidant capabilities as well as the nutritional value of brewer's grains. This is very interesting as using the process described here a "less valuable" waste could potentially be turned into highly valuable and protective ingredients. I would recommend acceptance after the following points are addressed by the authors.

1. The quality of the figures is very poor. Please provide Figures with higher quality.

2. In the fermentation/enzymatic hydrolysis experiments, the authors used DI water as medium. Given that enzymes are highly pH-sensitive, how did the authors ensure that there is no loss of activity during the process. Why not use a buffer in place of DI water.

3. Fig. 9 shows that in the absence of fermentation, the antioxidant activity of BGE sample is comparable to that of the control sample (CON) even though well-known biodegradation enzymes such as cellulase are present. Is this outcome due to lack of degradation or to formation degradation products with no antioxidant activity?

Author Response

Dear Reviewer,

Regards,

Round 2

Reviewer 1 Report

The authors have respond to all my comments.

The manuscript has been improved

Reviewer 2 Report

The paper “Integrated Metabolomics and Metagenomics Unveiled Biomarkers of Antioxidant Potential in Fermented Brewer’s Grains” has been revised according to the reviewer’s comments. The manuscript is acceptable for publication in this form.

The resolution of figures has been increased. The manuscript is acceptable for publication.